# TAxonomy of Self-reported Sedentary behaviour Tools (TASST) framework for development, comparison and evaluation of self-report tools: content analysis and systematic review

PM Dall,[1] EH Coulter,[2] CF Fitzsimons,[3] DA Skelton,[1] SFM Chastin,[1]
on behalf of the Seniors USP Team

[1]Institute for Applied Health Research, Glasgow Caledonian University, Glasgow, UK
[2]Department of Nursing and Healthcare, Glasgow University, Glasgow, UK
[3]Sport, Physical Education and Health Sciences Institute, Edinburgh University, Edinburgh, UK

**Correspondence to**
Dr P M Dall;
philippa.dall@gcu.ac.uk

## ABSTRACT

**Objective:** Sedentary behaviour (SB) has distinct deleterious health outcomes, yet there is no consensus on best practice for measurement. This study aimed to identify the optimal self-report tool for population surveillance of SB, using a systematic framework.

**Design:** A framework, TAxonomy of Self-reported Sedentary behaviour Tools (TASST), consisting of four domains (type of assessment, recall period, temporal unit and assessment period), was developed based on a systematic inventory of existing tools. The inventory was achieved through a systematic review of studies reporting SB and tracing back to the original description. A systematic review of the accuracy and sensitivity to change of these tools was then mapped against TASST domains.

**Data sources:** Systematic searches were conducted via EBSCO, reference lists and expert opinion.

**Eligibility criteria for selecting studies:** The inventory included tools measuring SB in adults that could be self-completed at one sitting, and excluded tools measuring SB in specific populations or contexts. The systematic review included studies reporting on the accuracy against an objective measure of SB and/or sensitivity to change of a tool in the inventory.

**Results:** The systematic review initially identified 32 distinct tools (141 questions), which were used to develop the TASST framework. Twenty-two studies evaluated accuracy and/or sensitivity to change representing only eight taxa. Assessing SB as a sum of behaviours and using a previous day recall were the most promising features of existing tools. Accuracy was poor for all existing tools, with underestimation and overestimation of SB. There was a lack of evidence about sensitivity to change.

**Conclusions:** Despite the limited evidence, mapping existing SB tools onto the TASST framework has enabled informed recommendations to be made about the most promising features for a surveillance tool, identified aspects on which future research and development of SB surveillance tools should focus.

**Trial registration number:** International prospective register of systematic reviews (PROPSPERO)/CRD42014009851.

### Strengths and limitations of this study

- A systematic approach was taken towards classifying self-reported measures of sedentary behaviour, allowing a structured approach to measurement in the future.
- An example of use of the framework is presented, mapping accuracy and sensitivity to change of self-reported sedentary behaviour (SB) measures on to the framework.
- Although designed to be generic, the TAxonomy of Self-reported Sedentary behaviour Tools framework was developed excluding tools measuring SB in specialised populations and contexts, for example, children or the workplace, and the framework may therefore not cover some aspects of these tools.
- There is the potential for a language bias, as full-text articles not in English were not included in the systematic reviews.

## BACKGROUND

Physical inactivity is currently at pandemic levels[1] and is a global public health concern. Sedentary behaviour (SB), an umbrella term for all waking time spent in non-exercising sitting or reclining postures[2 3] such as sitting during work, motorised transport or watching TV, is the largest contributor to inactivity.[4 5] Higher levels of SB have been associated with poor physical and mental health, increased risk of chronic disease and less successful ageing.[6–9] Consequently, several countries, including the UK, have issued recommendations to reduce SB at all ages as part of their national physical activity guidelines.[10] Population surveillance is urgently needed to monitor the impact of such policy, track changes in SB over time and to evaluate public health interventions

targeting SB. In order to provide effective surveillance on which to base future policy decisions, such surveillance tools should be accurate (provide a true measure of the actual amount of SB in a population) and sensitive to change (provide the true difference in SB between two measurement time points).[11]

Objective body-worn sensors, that measure posture, demonstrate good accuracy for measuring total duration of SB against the gold standard of direct observation,[12] but they are expensive and challenging to use for population surveillance. Self-report tools provide a pragmatic choice for population surveillance and have the potential to provide context-rich information, useful for intervention development.[13] To date, surveys assessing SB have predominantly used self-report tools,[14] which are generally adapted from tools not specifically designed to measure that behaviour (eg, tools designed to measure physical activity),[15] and which have not been evaluated for population surveillance purposes.[14] No framework currently exists with which to describe and compare SB self-report tools, meaning there is currently no way of systematically selecting an appropriate tool. A previous systematic review of the measurement characteristics of self-report tools measuring SB, reported acceptable to good reliability but low to moderate correlation with a (non-gold standard) criterion measure.[13] This suggests that self-report measures of SB are acceptable tools to establish epidemiological evidence of an association between SB and health.[13] However, it is possible that the scale of the problem may be vastly underestimated, as differences of 2–4 hours per day (~20% of SB) have been reported between self-report and objective tools.[16]

The primary aim of this study was to identify, in a systematic manner, the optimal self-report tool to measure SB for use in population surveillance. Although self-report SB tools can and will be used in other areas of research, this study focussed on population surveillance as an area that is crucial to the development of public health policy. To fulfil the primary aim, a framework was created to describe the features of self-report tools measuring SB, the TAxonomy of Self-report Sedentary behaviour Tools (TASST). A systematic inventory of existing self-report tools to measure SB was mapped onto TASST, and the measurement characteristics of these tools, focussing on accuracy and sensitivity to change, were evaluated, with explicit reference to the domains of the taxonomy framework.

## METHODS
The study protocol (International prospective register of systematic reviews (PROSPERO) CRD42014009851), was conducted in three phases. In phase 1 an exhaustive inventory of self-report tools to measure SB in adults and older adults was established using a structured search protocol. Phase 2 was the development of a taxonomy based on content analysis of the items and questions in the tools. In phase 3, a systematic literature review of the measurement characteristics of the tools in the inventory was conducted and mapped onto the taxonomy.

### Phase 1: systematic inventory of self-report tools
The aim of the systematic inventory was to compile an exhaustive list of self-report tools which could be used to measure SB in adults (≥18 years) and older adults (≥60 years). Since the aim was to identify tools and not to identify articles, this stage does not have the same methodology as a systematic literature review. A literature search was conducted in October 2013 (updated November 2016), for articles reporting SB as an outcome measure. From this review, a list of self-report tools which measured SB was compiled. References lists were reviewed and experts consulted to identify any additional tools to include in the inventory. The inventory then was consolidated to amalgamate tools referred to by different names, and to trace back to the original version. Articles which added significant new questions to tools were included as a separate tool. We defined significant new questions to be at least one question which added or changed the type of SB or the time period considered by the tool. Changes in phrasing of the question were not considered sufficient to be considered as a separate tool. Tools used in a single study and those without names/acronyms were included as separate tools.

To be included in the inventory, tools had to: be suitable for use for large-scale population studies of adults or older adults, including being suitable for self-completion by the respondent at a single point in time (a pragmatic requirement to minimise participant burden); and measure SB or a proxy measure of SB (eg, TV viewing). Although there is great interest in the SB across many populations and contexts, for pragmatic purposes, initial taxonomy development was limited to a core of self-report tools widely applicable to the general adult population. Therefore, tools were excluded from the inventory: if they were designed specifically to assess SB in children or other specialised populations (eg, medical conditions); if they were designed specifically to assess SB in a specialised context (eg, workplace or care settings); if continuous reporting over extended periods of time required (eg, diaries or time-use surveys) or if significant interviewer interactions were required. Self-report tools that could be administered by telephone or interview were not automatically excluded; however, tools such as the previous day recall (PDR),[17] in which the interviewer works through lists of several hundred items, were excluded.

### Phase 2: development of a taxonomy
Only tools identified in the initial search were used to develop the taxonomy. The original text was extracted for each question relating to SB in each of the self-report tools identified in the inventory. Content analysis was conducted on the text to extract all of the attributes

in the questions that were used to describe and constrain what aspect of SB was measured by that question. For example, in the question 'During the last 7 days, how much time did you usually spend sitting on a week day?', attributes extracted relating to the measurement of SB would be 'during the last 7 days', 'time spent sitting' and 'on a week day'. Attributes were then grouped into mutually exclusive domains covering similar aspects of measurement, and categories within those domains were defined iteratively. A new category was created each time a tool did not fit within an existing category. The full taxonomy was then assembled and streamlined by merging categories with overlapping meaning. Finally, consideration was given to potential future developments of self-report tools to measure SB, such as the growing interest in the pattern of accumulation of SB, by adding any categories to the taxonomy considered useful in the future. The resulting taxonomy was then tested by ensuring that all tools could be classified similarly by two independent researchers and that the taxonomy fully defined the tool.

## Phase 3: systematic review of measurement characteristics

Finally, a systematic literature search in relevant health databases was conducted in December 2014 (updated November 2016) via EBSCO host. The search combined the name of the tool including variants and acronyms (except where the acronym was also a common word, eg, Past-day Adults Sedentary Time questionnaire (PAST), Measuring Older adults' Sedentary Time questionnaire (MOST)), with search terms relating to measurement characteristics (valid* /reliab* /repons* /sensitiv* /calibrat* /accura* /agreement /psychometric* /clinimetric* /"measurement characteristics" /reliability and validity (Medical Subject Headings (MeSH))). Articles were included only if they reported in English on the accuracy of a tool in the inventory against an objective criterion measure of SB, and/or sensitivity to change. Although articles were only included in the review if they assessed accuracy or sensitivity to change, the search terms included a wide range of psychometric properties in order to maximise the chances of finding eligible articles.

Exclusion by title, then abstract, then full text was conducted by two researchers from a pool of five (PMD, EHC, CFF, SFMC and CL). In the case of disagreement, the article was carried forward in to the next round, or at full-text stage a third researcher was consulted to ensure consensus. Data (tool, criterion, population, statistical analysis, accuracy of sedentary behaviour and sensitivity to change of sedentary behaviour) were extracted and quality was assessed independently by two researchers from a pool of three (PMD, CFF, SFMC). Disagreements were resolved by discussion. Quality was assessed using QualSyst,[18] modified to include an additional item for the criterion measure. As per the QualSyst guidelines, the quality score for the article

(range 0–1) was used to identify common methodological strengths and flaws, rather than as an objective representation of high/low quality. Accuracy and sensitivity to change extracted from included articles were reported for tools in relation to the TASST taxonomy.

## RESULTS
### Inventory
The systematic inventory identified 37 distinct self-report tools used to measure SB in adults and older adults, 32 of which were identified in the initial search and used to form the taxonomy (table 1). The International Physical Activity Questionnaire (IPAQ) was originally developed with four different versions, which were included separately in the inventory (combinations of the long and short versions, and last 7 days and usual week recall). The 45 and Up Study asked different questions in its baseline and follow-up questionnaires, which have been included as separate tools. Three tools, termed 'modified' versions, were included where questions had been added or modified to the original tool (European Prospective Investigation of Cancer (EPIC)-Norfolk Physical Activity Questionnaire (EPAQ2), National Health and Nutrition Examination Survey (NHANES) and IPAQ-L, representing a 5th version of the IPAQ in the inventory), and were considered to form a substantially different version. Some tools identified were used in only a single study, and these were included in the inventory, referred to by the study name. The 32 tools in the original inventory comprised of 141 individual questions, consisting of between one and 20 questions per tool. An evaluation of the content of these individual items formed the basis of the TASST taxonomy.

### TAxonomy for Self-report Sedentary Behaviour Tools
The taxonomy derived from the inventory of self-report tools to measure SB (figure 1) comprises of four domains, which characterise different aspects of the tool: type of assessment, recall period, temporal unit and assessment period. All four aspects are required to describe the tool. Within each aspect, the taxonomy functions as a tree, meaning you can identify a single end point (taxon) which fully describes each question in a tool.

The type of assessment domain of the taxonomy covers the way that the outcome of time spent in SB is derived from the tool. Tools can ask about a single aspect of SB (1.1 single item), or a composite aspect (1.2 composite). Tools using a single item of assessment will generate all of their information about SB within the relevant period of assessment in a single question. That single item can either ask about sitting time directly (1.1.1 direct measure) or it can ask about a single behaviour related to SB which is then used as a proxy measure of SB duration (1.1.2 proxy measure). Composite items of assessment ask multiple questions about several aspects of SB for the same period of

**Table 1** Tools measuring SB for population surveillance identified in the inventory

| Acronym | Name of tool/study | Key reference |
|---|---|---|
| 45Up-B | 45 and Up Study, baseline questionnaire | [19] |
| 45Up-F | 45 and Up Study, follow-up questionnaire | [19] |
| ACS2 | American Cancer Society, Cancer Prevention Study cohort II | [20] |
| ALTS | Australian Leisure Time Sitting questionnaire | [21] |
| AusDiab | The Australian Diabetes Obesity and Lifestyle study | [22] |
| CCHS | Canadian Community Health Survey | [23] |
| CFS | Canadian Fitness Survey | [24] |
| CHAMPS | Community Health Activities Model Program for Seniors physical activity questionnaire | [15] |
| ELSA | English Longitudinal Study of Ageing | [25] |
| EPAQ2 | European Prospective Investigation of Cancer (EPIC)-Norfolk Physical Activity Questionnaire | [26] |
| mod EQPAQ2 | modified version of the EPIC-Norfolk Physical Activity Questionnaire | [27] |
| GPAQ | Global Physical Activity Questionnaire | [28] |
| HSE | Health Survey for England | [29] |
| HUNT3 | Nord-Trøndelag Health Study 3 | [30] |
| IPAQ-L l7d | International Physical Activity Questionnaire, Long version, last 7 days | [31] |
| IPAQ-L uw | International Physical Activity Questionnaire, Long version, usual week | [31] |
| mod IPAQ-L | modified version of the International Physical Activity Questionnaire, Long version | [32] |
| IPAQ-S l7d | International Physical Activity Questionnaire, Short version, last 7 days | [31] |
| IPAQ-S uw | International Physical Activity Questionnaire, Short version, usual week | [31] |
| LASA | Longitudinal Ageing Study Amsterdam | [33] |
| MLTPAQ | Minnesota Leisure Time Physical Activity Questionnaire | [34] |
| MOST | Measuring Older adults' Sedentary Time questionnaire | [35] |
| NHANES | National Health and Nutrition Examination Survey | [36] |
| mod NHANES | modified version of the National Health and Nutrition Examination Survey | [37] |
| NHS2 | Nurses Health Survey II | [38] |
| NIH-AARP DHS | National Institutes of Health—American Association of Retired Persons (NIH-AARP) Diet and Health Survey | [39] |
| NSWPAS | New South Wales Physical Activity Survey | [40] |
| PASE | Physical Activity Scale for the Elderly | [41] |
| PAST | Past-day Adults Sedentary Time questionnaire | [42] |
| PAST-U* | Past-day Adults Sedentary Time questionnaire—University version | [43] |
| PCSpa | prospective cohort study (Spain) | [44] |
| SBQ | Sedentary Behaviour Questionnaire | [45] |
| SHS | Scottish Health Survey | [46] |
| SIT-Q* | SIT-Q | [47] |
| SIT-Q-7d* | past seven day version of the SIT-Q | [48] |
| STAR-Q* | Sedentary Time and Reporting Questionnaire | [49] |
| STAQ* | Sedentary, Transportation and Activity Questionnaire | [50] |

Acronym: the commonly used acronym of the tool, or the short identifier adopted for this article. Name of tool: the name of the tool, or the name of the single study using these questions/tool. Key reference: references provided here are not exhaustive, but refer either to an early or well-cited description of the tool, or the study in which the tool was used or adapted. Tools marked with an asterisk (*) were identified in the updated search, and were not used to create the taxonomy.
SB, sedentary behaviour.

assessment. One form of composite item would be to ask about the pattern (ie, frequency and timing) of SB accumulated throughout the recall period (1.2.1 pattern). However, the most common form of composite item is created as a sum (1.2.2 sum) of the time spent in SB in a range of different activities or situations. The sum can be formed from questions asking about specific behaviours (1.2.2.1), activities such as TV viewing, hobbies and talking with friends, or they can be based on domains (1.2.2.2), locations or situations where you can sit, such as at home, for transport and at work.

The recall period is total time over which the respondent is asked to consider their SB when answering the questions. The recall period can be anchored to the present time in which case it refers to a specific length of time prior to now, for example, yesterday (2.1 previous day), last week (2.2 previous week) or a longer period such as the last month or year (2.3 longer). The recall period can also be unanchored (2.4), in which case the respondent is not asked about a specific period but is asked about a general period of time, for example, asking about SB in a typical week.

The temporal unit is the duration within the recall period that a respondent is asked to report their SB for. For example, in the question 'on a typical day last week, how long did you sit?' the recall period is the previous

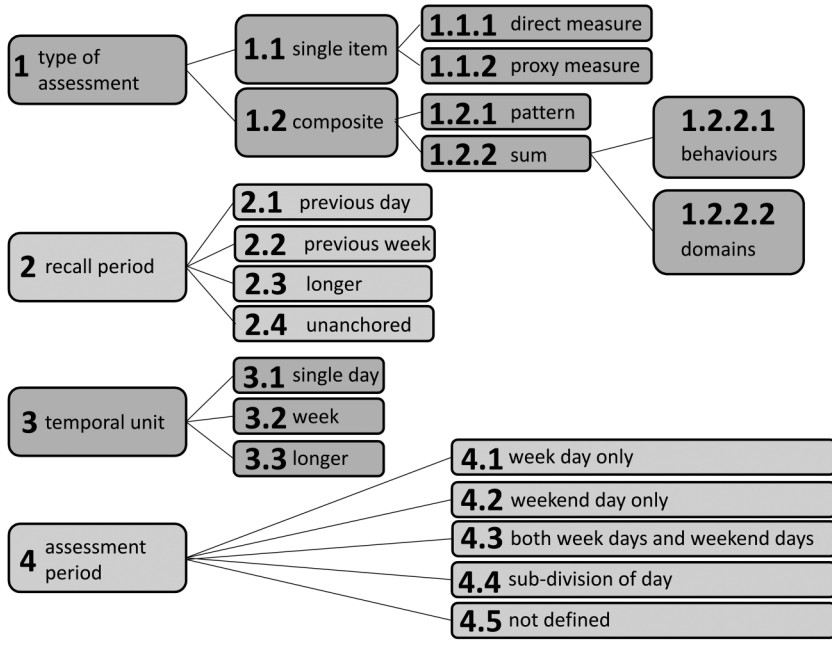

**Figure 1** TAxonomy of Self-reported Sedentary behaviour Tools (TASST).

week, but the temporal unit is a day. Within the taxonomy, the temporal units may be a day (3.1), a week (3.2) or longer (3.3). Within a particular recall period, it is possible to have any temporal unit that is of identical or shorter duration than the recall period.

The period of assessment is completed by identifying any specific restrictions that are placed on the type of temporal unit recalled. The categories within the assessment period domain clarify whether a respondent is asked questions regarding a particular type of day, for example, only about week days (4.1), only weekend days (4.2) or is asked about weekdays and weekend days in separate questions (4.3 both). Additionally, the assessment period domain can identify if a respondent is asked about particular subdivisions of the day (4.4) in separate questions, for example, time spent sitting before 6 pm. The final taxon in the assessment period is termed 'not defined' (4.5), this represents the situation where a respondent is asked about all temporal units (eg, days) within the recall period (eg, last week) without any specific distinction being made between them. It is a global category, which usually represents a decision not to separate out these categories, as opposed to a failure to define this domain.

### Mapping the inventory on to the taxonomy

The 37 tools identified in the inventory were mapped against the TASST taxonomy (table 2). Approximately half of the tools in the inventory (n=17) used a single item of assessment, 13 used a direct measure and 7 used a proxy measure. Three tools (45 and Up Study, baseline questionnaire (45Up-B), The Australian Diabetes Obesity and Lifestyle study (AusDiab) and National Institutes of Health—American Association of Retired Persons (NIH-AARP) Diet and Health Survey (NIH-AARP DHS)) asked single-item questions about a direct measure and a proxy measure, but not in a manner in which they could be used as a sum, and have therefore been included in the count for both taxa. Proxy measures were predominantly based on TV viewing (n=5). Twenty tools used composite assessment, all of which used a sum as that composite item. The vast majority of sums were formed from questions asking about different behaviours (n=19), with only one sum formed from questions asking about different domains. The tools using a sum of behaviours generally included the common proxy measures of TV viewing (n=19) and computer use (n=17) within the sum. Many tools included questions for behaviours based on leisure pursuits (n=14), in social contexts (n=9), and during transportation (n=13). Often several behaviours of each type were considered in separate questions (eg, asking about time sitting while reading separately from time spent sitting listening to music). Questions based on time working were included in 10 tools, but were explicitly excluded in four tools. Less frequently, tools included questions based on rest (n=5), or used an 'other' category to cover circumstances not explicit within the questions (n=7).

A little under half of the tools in the inventory used an unanchored recall period (n=15), 9 used a previous week recall period and 11 used a longer recall period. Only two tools (PAST, Past-day Adults Sedentary Time questionnaire—University version (PAST-U)) in the inventory used a PDR period. The majority of tools used a temporal unit of a day (n=32), with five (Australian Leisure Time Sitting questionnaire (ALTS), Canadian Community Health Survey (CCHS), Community Health Activities Model Program for Seniors physical activity questionnaire (CHAMPS), MOST, Nurses Health Survey II (NHS2)) using a temporal unit of a week. A single question within the EPAQ2 questionnaire was based on

**Table 2** Mapping of the tools measuring SB identified in the inventory onto the TASST taxonomy

| Taxonomy item | | N | Tools | Accuracy | Sensitivity to change |
|---|---|---|---|---|---|
| 1 | *Type of assessment* | | | | |
| 1.1 | Single item | 17 | | **Underestimate with large systematic and a random error** | + |
| 1.1.1 | Direct measure | 13 | 45Up-B; ACS2; AusDiab; CFS; GPAQ; HUNT3; IPAQ-L l7d; IPAQ-L uw; IPAQ-S l7d; IPAQ-S uw; NIH-AARP DHS; PASE; PCSPa | | |
| 1.1.2 | Proxy measure | 7 | 45Up-B; AusDiab; ELSA; MLTPAQ; NIH-AARP DHS; NSWPAS; SHS | | |
| 1.2 | Composite item | 20 | | **Smaller systematic error but there is a potential to overestimate** | + |
| 1.2.1 | Pattern | 0 | | | |
| 1.2.2 | Sum | | | | |
| 1.2.2.1 | Behaviours | 19 | 45Up-F; ALTS; CCHS; CHAMPS; EPAQ2; mod EPAQ2; HSE; mod IPAQ-L; LASA; MOST; NHANES; mod NHANES; PAST; PAST-U; SBQ; SIT-Q; SIT-Q-7d; STAR-Q; STAQ | | |
| 1.2.2.2 | Domains | 1 | NHS2 | | |
| 2 | *Recall period* | | | | |
| 2.1 | Previous day | 2 | PAST; PAST-U | **+** | − |
| 2.2 | Previous week | 9 | 45Up-F; ALTS; AusDiab; IPAQ-L l7d; IPAQ-S l7d; mod IPAQ-L; MOST; PASE; SIT-Q-7d | − | + |
| 2.3 | Longer | 11 | ACS2; CCHS; CHAMPS; EPAQ2; mod EPAQ2; HSE; NHANES; NIH-AARP DHS; SIT-Q; STAR-Q; STAQ | − | − |
| 2.4 | Unanchored | 15 | 45Up-B; CFS; ELSA; GPAQ; HUNT3; IPAQ-L uw; IPAQ-S uw; LASA; MLTPAQ; mod NHANES; NHS2; NSWPAS; PCSpa; SBQ; SHS | | |
| 3 | *Temporal unit* | | | | |
| 3.1 | Day | 32 | 45Up-B; 45Up-F; ACS2; AusDiab; CFS; ELSA; EPAQ2; mod EPAQ2; GPAQ; HSE; HUNT3; IPAQ-L l7d; IPAQ-L uw; IPAQ-S l7d; IPAQ-S uw; mod IPAQ-L; LASA; MLTPAQ; NHANES; mod NHANES; NIH-AARP DHS; NSWPAS; PASE; PAST; PAST-U; PCSpa; SBQ; SHS; SIT-Q; SIT-Q-7d; STAR-Q; STAQ | **+** | + |
| 3.2 | Week | 5 | ALTS; CCHS; CHAMPS; MOST; NHS2 | − | − |
| 3.3 | Longer | 0 | | − | − |
| 4 | *Assessment period* | | | | |
| 4.1 | Weekdays only | 2 | IPAQ-S l7d; IPAQ-S uw | − | + |
| 4.2 | Weekend days only | 0 | | − | − |
| 4.3 | Both weekdays and weekend days | 14 | 45Up-F; AusDiab; ELSA; HSE; IPAQ-L l7d; IPAQ-L uw; mod IPAQ-L; LASA; NSWPAS; PCSpa; SBQ; SHS; SIT-Q-7d; STAQ | **+** | − |
| 4.4 | Subdivision of the day | 1 | EPAQ2 | **+** | − |
| 4.5 | Not defined | 21 | 45Up-B; ACS2; ALTS; CCHS; CFS; CHAMPS; EPAQ2; mod EPAQ2; GPAQ; HUNT3; MLTPAQ; MOST; NHS2; NHANES; mod NHANES; NIH-AARP DHS; PASE; PAST; PAST-U; SIT-Q; STAR-Q | **Better for older adults** | + |

Full names for the acronyms reported in the tools column can be found in table 1. Recommendations in bold are backed by evidence from the systematic review. Recommendations which are not bold are theoretical but no evidence could be found in the literature; '+' represents a positive attribute; '−' a negative attribute.
SB, sedentary behaviour.

a temporal unit longer than a week, but the other three questions in that tool were based on a temporal unit of a day. Just over half the tools (n=21) did not define specific days or time periods in their questions, but asked about the temporal unit within the recall period as a single entity. A total of 16 tools used questions specifically referring to week or weekend days, 14 asking about week and weekend days, while 2 asked only about week days. Only one tool (EPAQ2) referred to specific subdivisions of the day in their questions.

## Systematic search for measurement characteristics

The systematic search returned 7221 references, and after removal of duplicate and assessment against exclusion criteria (>99% agreement between reviewers), a total of 22 studies were included in the review (figure 2, table 3).

## Criterion measure

None of the studies tested the accuracy of the tool against direct observation. Only five studies[16 42 43 48 63] used a postural sensor that actually measures sitting time objectively (activPAL), the other 17 used an accelerometer built to measure low movement as a criterion measure (ActiGraph, actiHeart).

## Statistical analysis

Accuracy and limits of agreement were usually derived from Bland and Altman plots. Sensitivity to change was defined differently in the two articles which reported this measurement characteristic; one used t-test statistics,[42] one used the Guyatt Index.[35]

## Study quality

Studies which scored highly for quality tended to be purposefully designed to test measurement characteristics, rather than secondary analysis of data collected for another purpose. The most common loss of quality was

due to the use of accelerometers which assess low movement (eg, ActiGraph) as a criterion measure, as this does not measure the primary aspect of the definition of SB (ie, posture). Another issue which lowered quality was the manipulation of the criterion measure without clear justification. For example, some studies manipulated the count threshold (used to define SB) or included only SB bouts longer than a particular duration without justification or solid rationale.

## Tools and measurement characteristics

Table 3 summarises the results reported by these studies, arranged per measurement tool and mapped against the relevant taxon. Very few of the existing tools to measure SB using self-report have actually been investigated for these measurement characteristics. Accuracy has been reported for 10 out of the 37 tools identified in the inventory (IPAQ-L l7d, IPAQ-S l7d, Global Physical Activity Questionnaire (GPAQ), MOST, CHAMPS, Longitudinal Ageing Study Amsterdam (LASA), PAST, PAST-U, Sedentary, Transportation and Activity Questionnaire (STAQ), SIT-Q-7d). The most tested tools were the IPAQ in its long form, last 7 days[16 51–55] and short form, last 7 days.[56–60] The SIT-Q-7d was tested in three studies,[48 63–64] and the CHAMPS was investigated in two studies.[15 62] Information for other tools, GPAQ,[61] LASA,[33] MOST,[35] PAST,[42] PAST-U[43] and STAQ,[50] come from single studies. Reports of sensitivity to change are only available for two tools, MOST[35] and PAST.[42]

## Taxa tested

The literature provides measurement characteristics information for eight distinct full taxa:
- ▶ 1.1.1/2.2/3.1/4.3 with six studies on IPAQ-L usual week (uw);
- ▶ 1.1.1/2.2/3.1/4.1 with five studies on IPAQ-S uw;
- ▶ 1.1.1/2.4/3.1/4.5 with one study on GPAQ;
- ▶ 1.2.2.1/2.1/3.1/4.5 with one study on PAST and one study on PAST-U;
- ▶ 1.2.2.1/2.2/3.1/4.3 with three studies on SIT-Q-7d;
- ▶ 1.2.2.1/2.2/3.2/4.5 with one study on MOST;
- ▶ 1.2.2.1/2.3/3.2/4.5 with two studies on CHAMPS;
- ▶ 1.2.2.1/2.4/3.1/4.3 with one study on LASA and one study on STAQ.

For the assessment type, there is information for direct measures via single item (1.1.1, 12 studies) and for composite sums of behaviours (1.2.2.1, 10 studies). However, there is no information for direct proxy measures (1.1.2). For recall period, there is information on all four possible categories (2.1 previous day, 2 studies; 2.2 previous week, 16 studies; 2.3 longer, 2 studies and 2.4 unanchored, 2 studies). The unanchored recall period (2.4), used by 40% of the tools in the inventory, is particularly under-represented with only two studies in the validation review. For temporal scale, there is mostly information for assessment at day scale (3.1, 20 studies) and only three studies for the temporal scale of a week (3.2). This is broadly representative of usage by tools in

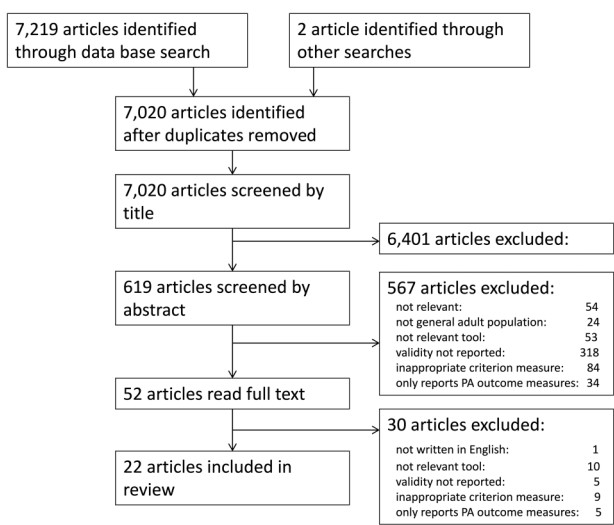

**Figure 2** Preferred Reporting Items for Systematic reviews and Meta-Analyses (PRISMA) diagram of the validation systematic review.

**Table 3** Measurement characteristics of tools measuring SB, presented by tool and taxon

| Tool | Taxon (refer to figure 1) | N | Population (Country) | Criterion measure (definition of SB) | QualSyst Score | Agreement (hours/day tool—criterion (limit of agreement) | Sensitivity to change | Ref |
|---|---|---|---|---|---|---|---|---|
| IPAQ-long l7d | 1.1.1/2.2/3.1/4.3 | 1508 | A & OA (Greenland) | actiHeart (<1.5MET) | 0.67 | −3.0 (not reported) for adults<br>−6.0 (not reported) for older adults | – | 51 |
| | | 542 | A (Netherlands) | Actigraph (<100 count/min) | 0.78 | −1.6 (−6.4 3.2) | – | 52 |
| | | 980 | A (Sweden) | Actigraph (<100 count/min) | 0.67 | +2.2 (−4.5 9.5) | – | 53 |
| | | 69 | A (UK) | activPAL (sitting/lying postures) | 0.78 | −2.2 (−7.2 3.7) | – | 16 |
| | | 317 | A (Chile) | Actigraph (<100 count/min) | 0.78 | −1.1 (−3.8 1.5) | – | 54 |
| | | 346 | A & OA (Switzerland) | Actigraph (<150 count/min) | 0.78 | −3.8 (−9.3 1.7) | – | 55 |
| IPAQ-short l7d | 1.1.1/2.2/3.1/4.1 | 1751 | A & OA (Norway) | Actigraph (<100 count/min) | 0.67 | −1.8 (not reported) for adults<br>+3.5 (not reported) for older adults | – | 56 |
| | | 144 | A (Nigeria) | Actigraph (<100 count/min) | 0.78 | −3.0 (−8.5 2.5) | – | 57 |
| | | 54 | OA (Sweden) | Actigraph (<100 count/min) | 0.56 | −1.5 (not reported) | – | 58 |
| | | 127 | OA (USA) | Actigraph (<50 count/min) | 0.72 | −4.4 (−10.0 −1.4) | – | 59 |
| | | 50 | A & OA (UK) | Actigraph (<50 count/min) | 0.72 | −0.5 (−1.9 0.8) | – | 60 |
| GPAQ | 1.1.1/2.4/3.1/4.5 | 62 | A (Saudi Arabia) | Actigraph (<100 count/min) | 0.67 | −3.3 (−9.7 3.1) | – | 61 |
| CHAMPS | 1.2.2.1/2.3/3.2/4.5 | 870 | OA (USA) | Actigraph (<100 count/min) | 0.72 | −6.8 (−10.6 2.4) | – | 15 |
| | | 58 | OA (USA) | Actigraph (<100 count/min) | 0.72 | −5.2 (not reported) | – | 62 |
| LASA | 1.2.2.1/2.4/3.1/4.3 | 83 | OA (Netherlands) | Actigraph (<100 count/min) | 0.78 | +0.2 for 10 items<br>−2.1 (−7.4 3.3) for 6 items | – | 33 |
| STAQ | | 88 | A (France) | Actigraph (<150 count/min) | 0.72 | −2.4 (−6.2 4.9) | – | 50 |
| PAST | 1.2.2.1/2.1/3.1/4.5 | 90 | A (Australia) | activPAL (sitting/lying postures) | 0.72 | −1.0 (− 5.6 3.8) | t-test was inconclusive | 42 |
| PAST-U | | 57 | A (Australia) | activPAL (sitting/lying postures) | 0.78 | 0.1 (−3.9 4.1) | – | 43 |
| SIT-Q-7d | 1.2.2.1/2.2/3.1/4.3 | 51 | A (Belgium) | activPAL (sitting/lying postures) | 0.72 | 1.0 (−4.8 8.2) for Belgian sample | – | 48 |
| | | 402 | A (UK) | actiHeart (<1.5MET) | | 0.4 (−6.9 8.6) for UK sample | | |
| | | 33 & 33 | A & OA (Belgium) | activPAL (sitting/lying postures) | 0.83 | 2.3 (only reported as a %)<br>0.3 (−8.9 0.7) for older adults | – | 63 |
| | | 442 | OA (Belgium) | Actigraph (<100 count/min) | 0.83 | 1.36 (−6.0 3.3) | – | 64 |
| MOST | 1.2.2.1/2.2/3.2/4.5 | 48 | OA (Australia) | Actigraph (<100 count/min) | 0.67 | −3.6 (−7.4 −0.2) | Guyatt Index 0.39 (0.47 for Actigraph) | 35 |

For tool acronyms see table 1.

A, adults; CHAMPS, Community Health Activities Model Program for Seniors physical activity questionnaire GPAQ, Global Physical Activity Questionnaire; IPAQ, International Physical Activity Questionnaire; LASA, Longitudinal Ageing Study Amsterdam; MOST, Measuring Older adults' Sedentary Time questionnaire; N, number of participants; OA, older adults; PAST, Past-day Adults Sedentary Time questionnaire; PAST-U, Past-day Adults Sedentary Time questionnaire—University version; Ref, reference; SB, sedentary behaviour; STAQ, Sedentary, Transportation and Activity Questionnaire; UK, UK; USA, USA of America.

the inventory. For assessment period, there is information for weekdays only (4.1, five studies) or both weekdays and weekend days (4.3, 11 studies) and for tools with the assessment period not defined (4.5, six studies). The assessment period not defined taxon (4.5), used by over half the tools in the inventory, is under-represented by these validation studies.

## Accuracy

Information for taxon 1.1.1/2.2/3.1/4.3 (IPAQ-L-l7d) is not equivocal. The majority of studies reported a large underestimation of total SB time ranging from 1.1 hours in adults[54] to 6 hours in older adults.[51] One study reported that tools in this taxon overestimate total SB time by 2.2 hours in adults.[53] It is clear that the systematic error on estimates of total SB time using tools from this taxon is likely to be very large (several hours/day). The random error is also likely to be very large as the limits of agreement reported were consistently very large. Information for taxon 1.1.1/2.2/3.1/4.1 (IPAQ-S-l7d) is a little more consistent for adults. Tools in this taxon seem to underestimate total SB time by 1.5 to 3 hours in adults. However, in older adults this was less clear with reports of underestimation by 4.4 hours[59] and overestimation by 3.5 hours.[56] In both populations the error and limits of agreement were large, but not as large as for the previous taxon.

Although not entirely consistent, tools reporting information from a single item as a direct measure of sitting (taxon 1.1.1) tended to underestimate sitting, with underestimation ranging from $-0.5$[60] to $-6.0$[51] hours per day. Within those tools, the IPAQ-S-l7d (reporting only for week days in the past week, taxa 2.2 and 4.1) tended to have better agreement than the IPAQ-L-l7d (reporting for both week and weekend days in the past week, taxa 2.2 and 4.3) and the GPAQ (reporting over a longer recall period with the assessment period not defined, taxa 2.4 and 4.5). Tools reporting on a sum of behaviours (taxon 1.2.2.1), were more likely to overestimate sitting than for the single-item direct measure (taxon 1.1.1). Tools which reported on a sum of behaviours over the past day or past week (taxa 1.2.2.1 & 2.1 or 2.2) tended to have the closest agreement with objective criterion measures with most studies reporting agreement between $-1.0$ and $+2.3$ hours per day. Tools which reported sum of behaviours over a longer (taxon 2.3) or unanchored (taxon 2.4) recall period or which had a temporal unit of a week (taxon 3.2) reported larger underestimates ($-2.1$ to $-6.8$ hours/day). In particular, the CHAMPS tool, reporting for a recall period of a year (taxon 2.3) with a temporal unit of a week (taxon 3.2), had the largest differences for any tool. However, there were only a few studies reporting on these aspects, and such conclusions are necessarily tentative. Regardless of level of agreement, limits of agreement were large for all tools.

## Sensitivity to change

There is almost no information about sensitivity to change. The two studies that assessed sensitivity to change[35 42] provided little tangible information. The results were either inconclusive,[42] or reported the Guyatt index against a criterion measure which does not measure sitting.[35] While the latter provided some indication that the tools' sensitivity to change was similar to that of an objective measure of low movement, it does not give a clear indication as to whether it is sensitive to a change in total SB time. Neither of these studies reported the minimal detectable change,[65] a metric which provides an easily interpretable value of the capacity of a tool to detect a change.

## DISCUSSION

A taxonomy (TASST) for the systematic description and comparison of self-reported measures of SB has been established. TASST provides a rigorous framework for informed choice, development and evaluation of self-report tools. This framework has been used to review the measurement characteristics of existing tools in order to identify the optimum tool for population surveillance. The available evidence about measurement characteristics essential for population surveillance, namely accuracy and responsiveness to change, was insufficient to ascertain which tool currently used in practice is best. Accuracy was poor for all existing tools, with under and overestimation of total time spent in SB and large limits of agreement. In addition, there is a complete lack of evidence about their sensitivity to change. Mapping available evidence onto the TASST framework has enabled informed recommendations to be made about the promising features for a surveillance tool, and identification of the aspects on which future research and development of SB surveillance tools should focus.

The use of a coherent and robust taxonomy (TASST) to systematically evaluate and compare the characteristics of measurement tools is the main strength of this study. However, in terms of accuracy and sensitivity to change, the current published evidence does not cover the entire taxonomy. Consequently, at present, only tentative recommendations can be provided. The taxonomy can be used, however, to identify gaps in current research and provide focussed guidance for future research and development. During the development of TASST, self-report tools which aimed to measure SB in specific populations (eg, children, those with arthritis) or specialised contexts (eg, workplace) were not considered. However, TASST is a generic framework, so tools specific to these populations may already be fully described by the taxonomy. For example, a question asking about time spent sitting at school which is specific to children, would be covered under the subdivision of the day assessment period (taxon 4.4). Another consequence of the exclusion criteria is that evidence on accuracy and sensitivity to change of tools specific to these populations was not mapped on the taxonomy. Therefore, the conclusions drawn from the measurement characteristics

in this study are only valid for adults and older adults. Future research should be conducted using the TASST taxonomy to map existing self-report tools covering those populations and contexts currently excluded from taxonomy development (such as children, schools or the workplace) to identify areas for development. In addition, this study has the general limitations common to most systematic reviews, that is, included articles were restricted to those written in English, articles and tools published after the date of search were not included and any relevant articles not identified during the search will have been excluded.

The current study is the first to clearly define and focus on the measurement characteristics required for population surveillance (accuracy and sensitivity to change). There is only one other systematic review reporting on the measurement characteristics of self-report tools to measure SB,[13] which concentrated on validity (assessed through rank correlation) and reliability, which are the measurement characteristics relevant to establishing associations between SB and health. In agreement with the previous review, we found that the major flaw of most validation studies was the use of an inadequate criterion measure. The choice of criterion measure depends on the purpose of the tool. While direct observation should be considered the gold standard, if the purpose is to assess total sedentary time, then accurate postural sensors should be adequate (eg, activPAL). In this review, only five out of 22 studies used an adequate criterion measure. Instead, many studies used an accelerometer which measures low levels of movement at the hip (eg, ActiGraph) as a criterion measure, but such tools do not measure SB directly and can misclassify standing as sitting.[12] Although it is possible that criterion measure may have provided a confounding effect on agreement (eg, tools assessing previous day recall period (taxon 2.1, PAST, PAST-U) were only assessed against the activPAL), no clear trend towards better or worse agreement with a particular type of criterion measure or ActiGraph cut-off was apparent.

Despite the incomplete nature of the evidence, TASST enables the identification of desirable characteristics of self-report tools to measure SB when used for population surveillance. First, tools assessing total SB time as a sum of behaviours (taxon 1.2.2.1; CHAMPS, LASA, MOST, PAST, PAST-U, SIT-Q-7d, STAQ) provided better accuracy than single-item direct measurement (taxon 1.1.1; IPAQ-L-l7d, IPAQ-S-l7 d and GPAQ) tools, especially when comparing tools with equivalent recall periods. However, this will be dependent on the behaviours or domains included within the sum, and whether they are exhaustive, consistent and mutually exclusive. Tools with a non-exhaustive sum will underestimate total time, for example, the LASA, found that a six-item sum provided a better correlation with SB across the sample, but that a 10-item sum was more accurate.[33] Conversely, tools which contain behaviours which might occur concurrently (such as watching TV and using a tablet computer) may lead to an overestimate in total SB time.[63] Second, tools using a previous day recall period (taxon 2.1, PAST, PAST-U) tended to provide better accuracy than those with longer recall periods (taxa 2.2, 2.3 and 2.4). This corroborates recent research on the validity of computerised survey systems which assess SB using a past-day recall period.[17 66] However, although tools using PDR may be more accurate, it is likely that their sensitivity to change will be less good due to the higher underlying variability in daily SB.[67]

Most tools currently used for population surveillance of SB systematically underestimate the amount of SB by 2–4 hours per day. Yet, self-report tools are still the most practical and economical means of population surveillance. Therefore, policymakers and clinicians should be aware that reports of population SB time are likely to be grossly underestimated, and should be cognisant of this fact when making decisions on implementing, developing and evaluating policy and public health interventions. In addition, policymakers and clinicians should be cautious in interpreting any reported difference in population SB time as a real change. The dearth of information about sensitivity to change of these tools means that we do not know the magnitude of change required to be certain that a change is real and not background variation. Moving forward, development of national and international surveillance systems should not be undertaken assuming that a tool is adequate because it has been used previously. Instead, investment should be made in research to evaluate the sensitivity to change and accuracy of tools to measure SB, paying attention to the potential trade-off between these two measurement characteristics. Such research should be carefully planned, to ensure that meaningful comparisons are investigated. The TASST taxonomy should be used as a useful framework to facilitate such a systematic approach.

**Acknowledgements** The named authors present the study on behalf of the Seniors – Understanding Sedentary Patterns (Seniors USP) Team, which comprises: DAS (PI), SFMC, Simon Cox, EHC, Iva Čukić, PMD, Ian Deary, Geoff Der, Manon Dontje, CFF, Catharine Gale, Jason Gill, Malcolm Granat, Cindy Gray, Carolyn Greig, Elaine Hindle, Karen Laird, Gillian Mead, Nanette Mutrie, Victoria Palmer, Ratko Radakovic, Naveed Sattar, Richard Shaw, John Starr, Sally Stewart, Sally Wyke. The authors would like to thank Calum Leask (CL) for his help in updating both of the searches in November 2016.

**Collaborators** Seniors USP Team, which comprises: DAS (PI), SFMC, Simon Cox, EHC, Iva Čukić, PMD, Ian Deary, Geoff Der, Manon Dontje, CFF, Catharine Gale, Jason Gill, Malcolm Granat, Cindy Gray, Carolyn Greig, Elaine Hindle, Karen Laird, Gillian Mead, Nanette Mutrie, Victoria Palmer, Ratko Radakovic, Naveed Sattar, Richard Shaw, John Starr, Sally Stewart, Sally Wyke.

**Contributors** SFMC and DAS conceived and supervised the study. PMD and SC designed the study. PMD, EHC, CFF and SFMC collected the data and performed the systematic review. PMD, CFF and SFMC analysed the data for the taxonomy. PMD and SFMC analysed the data from the systematic review. All authors interpreted the data and critically revised the manuscript for important intellectual content.

**Funding** The study was performed in preparation for the Seniors USP: Understanding Sedentary Patterns project, which was supported by the Medical Research Council [MR/K025023/1].

**Competing interests** None declared.

**Provenance and peer review** Not commissioned; externally peer reviewed.

**Data sharing statement** Data can be obtained from the corresponding author on request.

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
