## [Reviewer comments · BMJ Open]

ARTICLE DETAILS

TITLE (PROVISIONAL)	The TAxonomy of Self-reported Sedentary behaviour Tools (TASST) framework for development, comparison and evaluation of self-report tools: content analysis and systematic review.
AUTHORS	Dall, Philippa; Coulter, Elaine; Fitzsimons, Claire F.; Skelton, Dawn; Chastin, Sebastian

VERSION 1 - REVIEW

REVIEWER	Charlotte Edwardson University of Leicester, UK
REVIEW RETURNED	18-Sep-2016

GENERAL COMMENTS	This study aimed to identify the optimal self-report tool for assessing sedentary behaviour in population surveillance research. The manuscript is well written and the study has been rigorously conducted. Overall, I feel that this would be an important addition to the literature even though the authors were unable to identify an optimal tool. However, I do feel that there are a few changes required or things that needs to be justified more clearly in order for it to be published. After reading this manuscript I was left wondering the following: why was this focused only on tools used in population surveillance research? Why was reliability information not extracted and reported for the tools (given that 'reliability' was used as a search term)? Why was the decision made to exclude tools if they assessed SB in a specialised context e.g., workplace? Given that there's a large body of sedentary behaviour research being conducted in the workplace I feel that this would be important to include (although this may be due to the focus on population surveillance research?). Other Minor Comments: Abstract: Line 26 - be specific that you are looking for the 'optimal self-report tool' Line 60 - you state tools measuring sedentary behaviour in specialised contexts e.g., children, were excluded. Children would be a population so a better example of context should be given Methods: Line 113 – it would be good to define what you have included as an adult and older adult, i.e., are adults ≥ 18 years and older adults ≥ 60 years Line 115 – the literature search and systematic review were conducted back in October 2013 and December 2013 respectively, this is a long time ago. The sedentary behaviour research field is moving so rapidly therefore I believe this needs updating.
--

	Furthermore, in the discussion it's stated that 'articles and tools published after the date of the search were not included' – this limitation is easily solved by updating the searches given that they were performed so long ago. Line 123 – how would you define 'suitable for use for large scale population studies'? Line 124 – I would suggest giving an example of what you mean by 'proxy measure of SB' Lines 145-146 – it would be helpful to give an example here Discussion Line 395 - Emphasise that only 2 studies used the activPAL as the criterion measure Line 402 – it would be helpful here to remind the reader which tool/s assessed a sum of behaviours Line 409 – again it would be helpful here to remind the reader which tool/s assessed SB over a previous day recall period
--	---

REVIEWER	Lauren Arundell Institute for Physical Activity and Nutrition, Deakin University, Australia
REVIEW RETURNED	27-Sep-2016

GENERAL COMMENTS	This paper examines existing sedentary behaviour measurement tools to create a taxonomy of self-report sedentary behaviour tools. This is then used to map the existing tools that have accuracy and sensitivity to change information in order to identify the best tool for population studies. The authors should be commended on including such a large amount of information in this paper. While there are a few minor comments, this is a timely paper that has the potential to make an important and novel contribution to the literature. Therefore, I provide the following recommendation: Minor revisions. Abstract  1. It would be clearer to move the four domains of the TASST in the results section to the methods/design section. 2. In the data sources section you could also include that reference lists and experts were consulted. 3. Spelling error in the fourth strengths/limitation – notm Background  1. As this manuscript focuses on sedentary behaviour and not physical inactivity, the references to physical inactivity should be reconsidered as the terms are not interchangeable. 2. The background provides a good rationale for limitation of the current literature and the ideal characteristics of a measurement tool. It would be good to also briefly justify why the eligibility criteria includes being a single time point measurement tool. Methods Well done on making this section sequential and clear for the reader despite the amount of information it contains. Some minor comments below:  1. Please include an indication of how 'significant new questions' were determined? By number or by topic? 2. What were the agreement rates between the reviewers in phase 3? Results Again, there is a large amount of information in this section yet it reads clearly, well done. A large discussion point that I think should be considered more is
--

	that the accuracy of the self-report measurement tools is being assessed against different criterion and the main one doesn't measure sitting but lack of movement. This limitation is briefly discussed in the third paragraph of the discussion section, however it would be good to elaborate on this, for example through comparison of the agreement between the studies using the same criterion and cut point.  1. When discussing mapping the inventory on to the taxonomy where possible it would be helpful to include the study names. For example, Three tools (x, y and z) asked single item questions.... 2. Line 174 says there are four versions of IPAQ however the table suggest five, please confirm and amend where necessary. 3. Lines 275 and 300, check full stops and reference position 4. Page 341: please include the taxon for the one study (ref 42) to be consistent with the rest of the section Well done and good luck.
--	--

VERSION 1 – AUTHOR RESPONSE

Reviewer: 1

Reviewer Name

Charlotte Edwardson

Institution and Country

University of Leicester, UK

Please state any competing interests or state 'None declared':

None declared

Please leave your comments for the authors below:

This study aimed to identify the optimal self-report tool for assessing sedentary behaviour in population surveillance research. The manuscript is well written and the study has been rigorously conducted. Overall, I feel that this would be an important addition to the literature even though the authors were unable to identify an optimal tool. However, I do feel that there are a few changes required or things that needs to be justified more clearly in order for it to be published.

After reading this manuscript I was left wondering the following

Why was this focused only on tools used in population surveillance research?

The study was focussed on population surveillance research because it represents a timely and important area of research, for which self-report tools of SB are likely to be required. Although we acknowledge that self-report tools of SB will be used in other research contexts, some aspects of measurement properties for those have been covered (for example an existing systematic review [13] of the reliability and correlation with criterion measures of Self-report SB tools, suitable for use in epidemiological research). We were aware when we started the work that we would be trying to synthesise a large number of self-report SB tools, and decided to restrict our initial attempts to tools used for a specific (but important) purpose (i.e. population surveillance). Although we feel we have made the case for population surveillance as an important area in the first paragraph of the background, we have added the following sentence to the aims of the study, to clarify selection of this area of focus. "Although self-report SB tools can and will be used in other areas of research, this

study focussed on population surveillance as an area that is crucial to the development of public health policy.” (lines 96-98)

Why was reliability information not extracted and reported for the tools (given that ‘reliability’ was used as a search term)?

We restricted the validation part of our review to accuracy and sensitivity to change as these are key measurement properties required by tools being used in population surveillance. However, there can at time be confusion in the literature regarding usage of particular phrases. While we were not attempting to identify studies which reported reliability, we did consider that studies may, for example, have reported both reliability and accuracy, but not used the term accuracy. We therefore wished to conduct a relatively wide search to identify eligible articles. We have included the following sentence in the text to clarify this. “Although articles were only included in the review if they assessed accuracy or sensitivity to change, the search terms included a wide range of psychometric properties in order to maximise the chances of finding eligible articles.” (lines 164-167)

Why was the decision made to exclude tools if they assessed SB in a specialised context e.g., workplace? Given that there’s a large body of sedentary behaviour research being conducted in the workplace I feel that this would be important to include (although this may be due to the focus on population surveillance research?).

When commencing this work, we were aware that there would be a large amount of information to try and aggregate into a framework, and decided to concentrate on a “core” set of tools which could be applied generally to the wider population. This required some difficult decisions to be made in terms of limiting the scope of the work. One of the more difficult decisions we made was to treat the workplace as a specific context. Clearly the workplace is of great importance, as a place where prolonged sedentary behaviour may be undertaken, often in circumstances where social or actual behaviour is constrained by occupational requirements. However, we wished to use a clearly defined set of criteria to limit the scope of the work, and the workplace (along with schools and hospitals) does represent a specific context, and tools reporting only on that context were excluded from the review. This has been reported in the limitations. Future work should perhaps consider what (if any) changes would be required to the TASST framework to incorporate self-report tools to measure workplace SB into the framework.

Other Minor Comments:

Abstract:

Line 26 - be specific that you are looking for the ‘optimal self-report tool’

Thank you, we have added the word ‘self-report’ (line 26)

Line 60 - you state tools measuring sedentary behaviour in specialised contexts e.g., children, were excluded. Children would be a population so a better example of context should be given

We have expanded the comment to include both populations and contexts, to more accurately describe the exclusions/limitations. (line 59)

Methods:

Line 113 – it would be good to define what you have included as an adult and older adult, i.e., are adults ≥ 18 years and older adults ≥ 60 years

We have added age definitions to this sentence. (line 115)

Line 115 – the literature search and systematic review were conducted back in October 2013 and December 2013 respectively, this is a long time ago. The sedentary behaviour research field is moving so rapidly therefore I believe this needs updating. Furthermore, in the discussion it's stated that 'articles and tools published after the date of the search were not included' – this limitation is easily solved by updating the searches given that they were performed so long ago.

Thank you for this comment, we have updated the reviews (November 2016), and found five additional tools for inclusion in the inventory, and eight additional studies for inclusion in the validation study. The methods, results and discussion have been updated accordingly. Predominantly, these updates consist of updating numbers and are distributed throughout the text. Changes relating to updates from the searches have been highlighted in yellow, to distinguish them from other changes in response to reviewer's comments. The main changes have been to add items to tables 1, 2 and 3, and to update the PRISMA diagram (figure 2). The additional tools were not used to revise the taxonomy, but have been added into reporting of mapping tools on to the taxonomy, and the validation results especially the second paragraph of the results on agreement (lines 357-372). The conclusions drawn from the validation review did not change substantially. References for the additional tools and validation studies have been added and references renumbered as appropriate.

Line 123 – how would you define 'suitable for use for large scale population studies'?

This statement was predominantly concerned with the practical aspects of using a tool on a large scale. We have rearranged the paragraph to clarify "be suitable for use for large scale population studies of adults or older adults, including being suitable for self-completion by the respondent at a single point in time (a pragmatic requirement to minimise participant burden);" (lines)

Line 124 – I would suggest giving an example of what you mean by 'proxy measure of SB'

We have added TV viewing as an example. (line 131)

Lines 145-146 – it would be helpful to give an example here

We have added the following example "such as the growing interest in the pattern of accumulation of sedentary behaviour", to the text, to illustrate the type of thinking we tried to do to future-proof the taxonomy. (line 152)

Discussion

Line 395 - Emphasise that only 2 studies used the activPAL as the criterion measure

We have added the following sentence "In this review, only five out of 22 studies used an adequate criterion measure." which reflects the revised figures after updating the review. (line 424)

Line 402 – it would be helpful here to remind the reader which tool/s assessed a sum of behaviours

Line 409 – again it would be helpful here to remind the reader which tool/s assessed SB over a previous day recall period

Tools representing sum of behaviours (1.2.2.1), single item direct question (1.1.1) and previous day recall period (2.1) have been listed after the taxa, as suggested. Lines 434-443.

Reviewer: 2

Reviewer Name
Lauren Arundell

Institution and Country
Institute for Physical Activity and Nutrition, Deakin University, Australia

Please state any competing interests or state 'None declared':
None declared

Please leave your comments for the authors below:

This paper examines existing sedentary behaviour measurement tools to create a taxonomy of self-report sedentary behaviour tools. This is then used to map the existing tools that have accuracy and sensitivity to change information in order to identify the best tool for population studies. The authors should be commended on including such a large amount of information in this paper. While there are a few minor comments, this is a timely paper that has the potential to make an important and novel contribution to the literature. Therefore, I provide the following recommendation: Minor revisions.

Abstract

1. It would be clearer to move the four domains of the TASST in the results section to the methods/design section.

This information has been moved to the design section (line 28)

2. In the data sources section you could also include that reference lists and experts were consulted.

These have been added to the data sources in the abstract (line 33), and some words have been removed elsewhere to remain within the abstract word count.

3. Spelling error in the fourth strengths/limitation – notm

This has been corrected, thank you for spotting it.

Background

1. As this manuscript focuses on sedentary behaviour and not physical inactivity, the references to physical inactivity should be reconsidered as the terms are not interchangeable.

We agree that the terms physical inactivity and sedentary behaviour are not interchangeable, and have been careful to use the correct term referred to in the research quoted. We also have a statement that sedentary behaviour as the largest contribution to physical inactivity. The question of whether we should be quoting an article reporting on physical inactivity, as a source in this article is valid. However, we feel that the reference [1] is an impactful demonstration of the potential scale of health issues relating to sedentary behaviour (as a component of physical inactivity), and would prefer to keep the reference.

2. The background provides a good rationale for limitation of the current literature and the ideal characteristics of a measurement tool. It would be good to also briefly justify why the eligibility criteria includes being a single time point measurement tool.

The eligibility criteria to be measured at a single time point relates back to the practical aspects of conduction large scale population surveillance. Most such studies require a pragmatic mixture of

amount of information gathered from a tool and participant burden. We have added the following phrase to clarify “(a pragmatic requirement for large scale studies to minimise participant burden)”. (line 130)

Methods

Well done on making this section sequential and clear for the reader despite the amount of information it contains. Some minor comments below:

1. Please include an indication of how ‘significant new questions’ were determined? By number or by topic?

We have added the following sentences to clarify this statement. “We defined significant new questions to be at least one question which added or changed the type of sedentary behaviour or the time period considered by the tool. Changes in phrasing of the question were not considered sufficient to be considered as a separate tool.” (lines 123-125)

2. What were the agreement rates between the reviewers in phase 3?

When screening by title and abstract, there were 7 disagreements (out of 7,020 articles reviewed), representing >99% agreement between reviewers. At full text, agreement was 100% between reviewers. A statement to that effect has been included in the results (line 284).

Results

Again, there is a large amount of information in this section yet it reads clearly, well done.

A large discussion point that I think should be considered more is that the accuracy of the self-report measurement tools is being assessed against different criterion and the main one doesn’t measure sitting but lack of movement. This limitation is briefly discussed in the third paragraph of the discussion section, however it would be good to elaborate on this, for example through comparison of the agreement between the studies using the same criterion and cut point.

Thank you for this comment; it is something we feel is important. However, although we have raised the issue of inappropriate criterion measures, and differences in ActiGraph criterion measure cut-points within the text, we found it difficult to identify any clear trend towards better or worse agreement depending on the type of criterion used. It is possible that some confounding between tool and criterion measure might be evident, for example the activPAL is the only criterion measure for tools with previous day recall period (taxon 2.1, PAST and PAST-U), both of which performed relatively well. However, when different criteria are used to assess agreement for the same tool, those assessed against activPAL show no tendency to provide a closer agreement (-2.2 hours per day for IPAQ-L-7d, compared to a range of -6.0 to +2.2). Both studies that used an ActiGraph cut-point of <50 count/min were assessing the IPAQ-S 7d, but these provide both the best and the worst agreement for the tool (the other three studies being assessed by an ActiGraph using a <100 count/min cut-off).

To address your comment within the text we have added the following sentence “Although it is possible that criterion measure may have provided a confounding effect on agreement (e.g. tools assessing previous day recall period (taxon 2.1, PAST, PAST-U) were only assessed against the activPAL, no clear trend towards better or worse agreement with a particular type of criterion measure or ActiGraph cut-off was apparent.” to the discussion. (lines 427-430).

1. When discussing mapping the inventory on to the taxonomy where possible it would be helpful to include the study names. For example, Three tools (x, y and z) asked single item questions....

We thank the reviewer for their comment to improve readability. Within the section on mapping the

taxonomy, there are some categories that have a large number of studies included (for example, a temporal unit of a day is used by 32 tools). The tools associated with each taxon are listed in table 2, and we felt that listing for the larger categories would be unhelpful. However, we appreciate it can be difficult/annoying to look up tool names for smaller groups, and especially when describing tools that are in two categories. We have added tool names to four of the smaller categories within these sections. (lines 251-280)

2. Line 174 says there are four versions of IPAQ however the table suggest five, please confirm and amend where necessary.

The IPAQ was originally constructed with four different versions. We also included a tool which was a modified version of one of those versions, leading to five different versions in the inventory table. The text has been amended to clarify this. Lines 185-191

3. Lines 275 and 300, check full stops and reference position 4.

Thank you for spotting this, we have corrected the sentences (now lines 290 and 315)

Page 341: please include the taxon for the one study (ref 42) to be consistent with the rest of the section

The paragraph that this comment referred to has been removed from the article, and rewritten in light of the additional articles included following the updated review (lines 357-372). For clarity, this text has been reviewed to make sure that taxa and descriptions are both provided.

Well done and good luck.

VERSION 2 – REVIEW

REVIEWER	Lauren Arundell Institute for Physical Activity and Nutrition, Deakin University, Australia.
REVIEW RETURNED	18-Jan-2017

GENERAL COMMENTS	Thank you for the opportunity to re-review your manuscript. The authors have done a good job of addressing the comments made by the reviewers and as such, have strengthened the manuscript. I therefore provide the following recommendation: accept. A minor comment is that it may be useful to briefly explain why tools specifically designed to assess sedentary behaviour in children were excluded (page 6, line 132). Using the TASST taxonomy to map children’s SB tools and tools measuring SBs in specific contexts also excluded in this manuscript (e.g. workplaces, schools) may be additional future research suggestions. Thank you for this timely paper which I believe will be useful to the sedentary behaviour research field.
---

VERSION 2 – AUTHOR RESPONSE

Reviewer: 2
Arundell, Lauren
Deakin University

Please leave your comments for the authors below Thank you for the opportunity to re-review your

manuscript. The authors have done a good job of addressing the comments made by the reviewers and as such, have strengthened the manuscript. I therefore provide the following recommendation: accept.

A minor comment is that it may be useful to briefly explain why tools specifically designed to assess sedentary behaviour in children were excluded (page 6, line 132).

Similar to our decision to exclude tools for specific contexts (such as the workplace, see response to reviewer 1 in the first set of revisions), the decision to exclude tools specifically designed to assess sedentary behaviour in children from our systematic inventory (and thus from development into the TASST taxonomy) was taken primarily to limit the scope of the original development to manageable proportions. In part this was informed by our informal knowledge both that a number of self-report tools specific to children existed, and that the physical activity and sedentary behaviour of children is often assessed separately to that of adults. However, the reasoning behind these decisions is not clear in the text of the article. Accordingly we have added the following sentence, "Although there is great interest in the sedentary behaviour across many populations and contexts, for pragmatic purposes initial taxonomy development was limited to a core of self-report tools widely applicable to the general adult population." (page 6, lines 131-133).

Using the TASST taxonomy to map children's SB tools and tools measuring SBs in specific contexts also excluded in this manuscript (e.g. workplaces, schools) may be additional future research suggestions.

Thank you for this suggestion. Such work was hinted at in our discussion, but we have now made an explicit statement. "Future research should be conducted using the TASST taxonomy to map existing self-report tools covering those populations and contexts currently excluded from taxonomy development (such as children, schools or the workplace) to identify areas for development." (page 24, lines 411-414).

Thank you for this timely paper which I believe will be useful to the sedentary behaviour research field.